# The Risk Factors for Musculoskeletal Injuries in Thoroughbred Racehorses in Queensland, Australia: How These Vary for Two-Year-Old and Older Horses and with Type of Injury

**DOI:** 10.3390/ani11020270

**Published:** 2021-01-21

**Authors:** Kylie L. Crawford, Anna Finnane, Clive J. C. Phillips, Ristan M. Greer, Solomon M. Woldeyohannes, Nigel R. Perkins, Lisa J. Kidd, Benjamin J. Ahern

**Affiliations:** 1School of Veterinary Science, The University of Queensland, Gatton 4343, Australia; s.woldeyohannes@uq.edu.au (S.M.W.); n.perkins1@uq.edu.au (N.R.P.); l.kidd@uq.edu.au (L.J.K.); b.ahern@uq.edu.au (B.J.A.); 2School of Public Health, The University of Queensland, Herston 4006, Australia; a.finnane@uq.edu.au; 3Curtin University Sustainability Policy (CUSP) Institute, Curtin University, Perth 6845, Australia; Clive.Phillips@curtin.edu.au; 4Torus Research, Bridgeman Downs 4035, Australia; rmg@torusresearch.com.au; 5School of Medicine, The University of Queensland, Herston 4006, Australia

**Keywords:** racehorse, thoroughbred, epidemiology, wastage, musculoskeletal injury

## Abstract

**Simple Summary:**

Musculoskeletal injuries (MSI) continue to affect Thoroughbred racehorses internationally, despite over thirty years of research into this problem. Studies of risk factors for musculoskeletal injuries report inconsistent findings. Consequently, developing training strategies to mitigate the risk of MSI is difficult. We identified factors associated with particularly high odds of injury in this population of racehorses. Two-year-old horses from primiparous mares (first foals born) are at increased odds of MSI, particularly dorsal metacarpal disease (“shinsoreness”). Two-year-old horses that have had a total preparation length of between 10 and 14 weeks also have increased odds of injury. Horses of all ages that travelled a total distance of 2.4–3.8 km (12–19 furlongs) at a gallop (faster than 15 m/s; 13 s/furlong; 900 m/min; 55 km/h) in the last four weeks and horses three years and older that travelled 3.0–4.8 km (15–24 furlongs) at three-quarter pace and above (faster than 13 m/s; 15 s/furlong; 800 m/min; 48 km/h) also have increased odds of injury. We recommend that these horses should be monitored closely for impending signs of injury. We also observed a non-linear relationship between high-speed exercise and musculoskeletal injuries. This highlights the importance of high-speed exercise to enable tissue adaptation to training. Finally, in some situations, increasing the number of days worked at a slow pace may be more effective for preventing MSI, if horses are perceived at a higher risk, than resting the horse altogether. Early identification of horses at increased risk of injury and appropriate intervention could substantially reduce the impact of musculoskeletal injuries in Thoroughbred racehorses.

**Abstract:**

Musculoskeletal injuries (MSI) continue to affect Thoroughbred racehorses internationally. There is a strong interest in developing training and management strategies to reduce their impact, however, studies of risk factors report inconsistent findings. Furthermore, many injuries and fatalities occur during training rather than during racing, yet most studies report racing data only. By combining racing and training data a larger exposure to risk factors and a larger number of musculoskeletal injuries are captured and the true effect of risk factors may be more accurately represented. Furthermore, modifications to reduce the impact of MSI are more readily implemented at the training level. Our study aimed to: (1) determine the risk factors for musculoskeletal injuries and whether these are different for two-year-old and older horses and (2) determine whether risk factors vary with type of injury. This was performed by repeating analyses by age category and injury type. Data from 202 cases and 202 matched controls were collected through weekly interviews with trainers and analysed using conditional logistic regression. Increasing dam parity significantly reduced the odds of injury in horses of all age groups because of the effect in two-year-old horses (odds ratio (OR) 0.08; 95% confidence interval (CI) 0.02, 0.36; *p* < 0.001). Increasing total preparation length is associated with higher odds of injury in horses of all ages (OR 5.56; 95% CI 1.59, 19.46; *p* = 0.01), but particularly in two-year-old horses (OR 8.05; 95% CI 1.92, 33.76; *p* = 0.004). Increasing number of days exercised at a slow pace decreased the odds of injury in horses of all ages (OR 0.09; 95% CI 0.03, 0.28; *p* < 0.001). The distance travelled at three-quarter pace and above (faster than 13 m/s; 15 s/furlong; 800 m/min; 48 km/h) and the total distance travelled at a gallop (faster than 15 m/s; 13 s/furlong; 900 m/min; 55 km/h) in the past four weeks significantly affected the odds of injury. There was a non-linear association between high-speed exercise and injury whereby the odds of injury initially increased and subsequently decreased as accumulated high-speed exercise distance increased. None of the racing career and performance indices affected the odds of injury. We identified horses in this population that have particularly high odds of injury. Two-year-old horses from primiparous mares are at increased odds of injury, particularly dorsal metacarpal disease. Two-year-old horses that have had a total preparation length of between 10 and 14 weeks also have increased odds of injury. Horses of all ages that travelled a total distance of 2.4–3.8 km (12–19 furlongs) at a gallop in the last four weeks and horses three years and older that travelled 3.0–4.8 km (15–24 furlongs) at three-quarter pace and above also have increased odds of injury. We recommend that these horses should be monitored closely for impending signs of injury. Increasing the number of days worked at a slow pace may be more effective for preventing injury, if horses are perceived at a higher risk, than resting the horse altogether. Early identification of horses at increased risk and appropriate intervention could substantially reduce the impact of musculoskeletal injuries in Thoroughbred racehorses.

## 1. Introduction

Musculoskeletal injuries (MSI) continue to affect Thoroughbred (TB) racehorses internationally despite substantial active research into this problem [1,2,3]. There are important ethical, welfare and economic consequences resulting from MSI. A principal issue is the serious injury and/or death of horses [4,5,6,7] and riders [8,9]. Musculoskeletal injuries are the most common cause of death, comprising over 70% of TB racehorse fatalities [10,11,12,13,14,15,16,17,18]. Previous studies report between 7% and 49% of race day MSI resulted in death of the horse [4,5,13,19,20,21,22,23]. Furthermore, riders are more likely to be seriously injured or killed when their horse suffers a MSI [8,9].

There has been, and continues to be, a strong interest in developing training and management strategies to reduce the impact of MSI. High-speed exercise history (HSEH) has been identified as an important risk factor for MSI due to the high loads and strains generated [24,25,26]. However, the association reported between HSEH and MSI is inconsistent. Some studies report that as HSEH increases [27,28,29,30,31,32,33,34,35,36] the risk of MSI increases, while others report it decreases [17,29,37,38,39,40,41,42,43] or does not change [17,19,44,45]. Furthermore, other studies report increasing HSEH has a curvilinear effect on the risk of MSI, whereby as HSEH continues to increase, the risk of MSI initially decreases, plateaus, then increases again [17,36,38,40]. Possible reasons for inconsistent reports include differences in case (outcome) definitions, study populations, geographical locations, the way HSEH is reported, the complex relationship between HSEH and MSI, or any combination of these factors.

Many MSI cases and fatalities occur during training rather than during racing [10,11,21,30,33,46]. Therefore, studies with a case definition of race day MSI will miss a large proportion of cases that occur during training. These studies will also not capture MSI cases that are not apparent on the day of racing and are discovered later [11,47]. Analysing combined training and racing data may be more beneficial than analysing racing data alone. By combining racing and training data a larger exposure to HSEH and a larger number of MSI are captured and the true effect of HSEH on MSI may be more accurately represented. Furthermore, modifications to reduce the impact of MSI are more readily implemented at the training level.

Studies report conflicting findings regarding the effect of age on MSI. Some studies report that two-year-old horses may be more susceptible to MSI due to their immaturity [48]. Other studies report age did not affect the incidence of MSI [7,17,49]. The inconsistent association between age and MSI may be due to variation in case definitions, study populations and geographical locations, or a combination of these factors. A recent prospective study found that the incidence and type of injuries varied between two-year-old and older horses [50]. Therefore, it is highly likely that the risk factors for MSI in two-year-old horses may be different than those for older horses. It is also likely that the risk factors may vary between types of MSI.

We address these data gaps through a case control study conducted over a 13-month period. Our aims were to: (1) determine the risk factors for MSI and whether these are different between two-year-old and older horses and (2) determine whether the risk factors vary according to type of MSI. The Thoroughbred racing industry attracts widespread media and public attention and therefore, it is vital that independent research is conducted to reduce the impact of these injuries.

## 2. Materials and Methods

### 2.1. Recruitment of Participants

This study was performed concurrently with a study investigating the incidence and type of MSI in Thoroughbred racehorses in South-East Queensland, Australia, which describes our recruiting process in detail [50]. Human and animal ethics approvals were obtained from the University of Queensland prior to commencement of this research (approval numbers 2017001248, SVS/384/17 respectively). The structure of Thoroughbred racing in Queensland, Australia is described in Figure A1 of the Appendix A. Briefly, there are eight regions, and each region is divided into Totalisator Agency Board (TAB) and non-TAB racing clubs. Trainers from the three metropolitan racetracks controlled by the Brisbane Racing Club (BRC) were invited to participate in the survey. These tracks (Eagle Farm, Doomben, and Deagon) are the major racetracks in South-East Queensland. Trainers exercised their horses at Eagle Farm, Doomben, Deagon, or both Doomben and Eagle Farm. Recruitment of horses was performed by recruiting trainers and enrolling all the horses under their care. The first author invited all licensed trainers at Eagle Farm, Doomben, and Deagon with three or more horses in work at the time of recruitment to participate in the study. We opted for a minimum of three horses to ensure trainer capacity to supply sufficient control horses.

### 2.2. Study Design

This study was a matched case-control study with cases and controls recruited prospectively over a 13-month (56 week) period. The training stables were monitored weekly and incident cases were identified. Controls were recruited concurrently with every incident case. Detailed training and exercise data were collected for both cases and controls through personal structured weekly interviews with participating trainers or their forepersons. Structured personal interviews enabled clarification of any inconsistencies observed, ensuring accurate and complete data collection. Details of the interview are described in Figure A2.

### 2.3. Data Collection

The data of interest were:(1)Cases of MSI and the diagnosis(2)One control matched to every case of MSI(3)Risk factors for MSI

#### 2.3.1. Case Selection

The outcome of interest was a MSI, defined as any clinically relevant injury to the musculoskeletal system, incorporating orthopaedic and soft tissue injuries which prevented the horse from training for seven days. A seven-day period was chosen as consistent with previous studies [11,51]. Our definition included any MSI that occurred whilst the horse was in training, whether the actual injury occurred during a race, training, or following an accident in the stable. We included osteochondritis dissecans, cervical stenotic myelopathy and other developmental orthopaedic conditions if the horse was in training, sound and later developed a clinical lameness or gait abnormality that prevented them from training. Musculoskeletal injuries were diagnosed by a veterinarian to avoid measurement and ascertainment bias. Horses in the study were under the close care of racetrack veterinarians registered in Queensland.

#### 2.3.2. Control Selection

One control was randomly selected from the same training stable when each incident case was identified. Controls were matched to cases by age. If there was no horse available that was the same age as the case, a control aged within 12 months of the case was randomly selected.

#### 2.3.3. Definitions of Speeds and Terminology

A training preparation was defined as the time in between rest periods that the horse was actively participating in race training exercise. We measured this time from when the horse entered the training stable (either after a rest period or for the first time) to when the horse left the stable and did not train for seven days. Slow gaits were defined as exercise less than 13 m/s (15 s/furlong; 800 m/min; 48 km/h). Water-walkers were defined as walking machines within a shallow swimming pool, whereby the horse completes a walking exercise in water. Treadmill exercise was defined as riderless exercise on a stationary device consisting of a continuous conveyor belt, whereby the exact speed of exercise could be controlled. Three-quarter pace and above was defined as exercise faster than 13 m/s (15 s/furlong; 800 m/min; 48 km/h). Galloping was defined as exercise faster than 15 m/s (13 s/furlong; 900 m/min; 55 km/h). Jump-outs were defined as unofficial barrier trials, whereby they are not broadcast, nor the horses’ performance recorded. Official trials were barrier trials that were broadcast and the results recorded in the official Racing Australia public database [52]. Races were races administered by Queensland Racing Integrity Commission and results were recorded in the official public Racing Australia database.

#### 2.3.4. Risk Factors for Musculoskeletal Injury

We investigated four broad categories of risk factors for MSI:(1)Population characteristics(2)General racing and training history(3)High-speed exercise history(4)Racing career and performance indices

##### Population Characteristics

Cases and controls were identified by both registered name and microchip number. Sex was recorded as female, entire male, or gelding. The dam ages and parities were obtained for each horse from the Australian Stud Book [52].

##### General Racing and Training History

Whether the horse had raced or not was obtained for each horse from the Racing Australia (RA) public database [52]. The length of time (in weeks) that each horse had completed in pre-training and training was obtained from the trainer or foreperson. The total preparation time was calculated by the sum of the pre-training and training periods. Daily exercise history was collected for the previous four weeks, as recommended by the Havemeyer workshop on the epidemiology of training and racing injuries [53]. This equated to four weeks of exercise history prior to MSI for cases, and four weeks prior to their matched case’s MSI for controls. The total number of days worked at slow gaits (less than 13 m/s; 15 s/furlong; 800 m/min; 48 km/h) and the number of days using water-walkers and treadmills were recorded.

##### High-Speed Exercise History

Daily exercise data for the previous four weeks were collected, from which we calculated eleven parameters of high-speed exercise. These parameters of high-speed exercise included:(1)The total number of days (events) exercised at “three-quarter pace and above” (faster than 13 m/s; 15 s/furlong; 800 m/min; 48 km/h)(2)The total distance (furlongs) travelled at “three-quarter pace and above”(3)The average distance per event at “three-quarter pace and above”(4)The total number of days (events) exercised at “gallop” (faster than 15 m/s; 13 s/furlong; 900 m/min; 55 km/h) during trackwork(5)The total distance (furlongs) travelled at “gallop” during trackwork(6)The average distance per event at “gallop” during trackwork(7)The total distance (furlongs) travelled in jump-outs (non-official trials)(8)The total distance (furlongs) travelled in official trials(9)The total distance (furlongs) travelled in races(10)The combined distance travelled for jump-outs, official trials, and races(11)The total distance travelled at “gallop” during trackwork, jump-outs, official trials, and races (furlongs)

The official trial and race data were cross-checked with the RA public database [52].

##### Racing Career and Performance Indices

Racing career and performance indices were obtained or calculated from each individual horse’s form on the RA public database [52]. These indices included the number of starts as a two-year-old, age at first race, length of the racing career in months, number of race starts, number of race starts per 12 months of racing, total race distance accumulated (the sum of the total distance of all races completed in 1 km (5 furlongs)/1 km units and the mean distance per start (furlongs). Performance indices included: the number and percentage of races won; the number and percentages of races the horse placed 1st, 2nd, or 3rd; the total prizemoney earned (1000 Australian dollars); and the prizemoney earned per race start (1000 Australian dollars). No horses raced outside of Australia for the duration of the study.

### 2.4. Power Calculations

Power calculations were based on the findings of Estberg (1996), whereby horses that accumulated 35 furlongs of race and timed-workout distance had a 3.9-fold increase in fatal fracture risk compared to those that accumulated 25 furlongs of race and timed-workout distance [31]. By plotting the number of matched pairs in increasing increments from 100 to 250, we determined that at least 180 matched pairs were required to detect an odds ratio of 2.0, at a power of 80%.

### 2.5. Data Analysis

Stata 15.1^®^ (Statacorp, College Station, TX, USA) was used to analyse data. Normality of continuous data was assessed using histograms. Normally distributed data were presented as mean and standard deviation. Non-normally distributed data were presented as median and interquartile range unless otherwise indicated. Categorical data were presented as numbers and percentages. Two-sample Wilcoxon rank-sum tests and Pearson’s Chi-squared tests were used to assess the differences in population characteristics, general racing and training history, HSEH, and racing career and performance indices between two-year-old and older horses. The effects of these variables on MSI were assessed using conditional logistic regression. This method of analysis corrects for misspecification due to the differences in training practices between trainers and resulting from age [54,55,56]. The assumption of linearity between continuous variables and the odds of musculoskeletal injury was checked by generating the predicted probabilities for each variable, and then categorizing them into four equal categories. The continuous variable was then divided into four categories and the actual odds of MSI for each category determined. The mean value of the predicted probability for each category was plotted against the actual odds of MSI for each category. If the association was not linear, the variable was analysed as a categorical variable [54].

Univariable conditional logistic regression was performed and variables were considered for inclusion in multivariable models if they were significant at α = 0.2 [57]. Pearson’s pairwise correlation tests were performed on all variables eligible for inclusion in the multivariable model. Multivariable analyses were performed using a backwards stepwise procedure to remove all non-significant variables from the model and obtain a parsimonious model. Significance was set at α = 0.05. The model was then repeated using logistic regression to generate predictive values and post-estimation diagnostics were performed to check for multicollinearity and misspecification by examining correlation matrices and predicted values [58]. The analyses were repeated for two-year-old and older horses.

#### Analysis by Type of Musculoskeletal Injury

The types of MSI observed in this population of horses have been described in detail elsewhere [50]. However, we were interested in whether the effect of significant risk factors for MSI varied according to injury type. The five broad categories of MSI that we analysed were:(1)Dorsal metacarpal disease(2)Carpal and fetlock injuries(3)Tendon and ligament injuries(4)Fractures(5)Other

Univariable conditional logistic regression analyses by type of injury were performed for variables significant in the main multivariable model for horses of all ages. Multivariable analyses by type of injury were not possible due to lack of power.

## 3. Results

There were 40 trainers (15 at Eagle Farm, 6 at Doomben, 12 at Deagon, and 7 at both Doomben and Eagle Farm) eligible to participate in our study. Of these, 27/40 (68%) gave informed consent (Figure A3).

Data were collected every week for 56 weeks (13 months) from November 2017 to December 2018 for 26/27 (96%) of trainers. One trainer (4%) only contributed six months of data before retiring from training.

### 3.1. Characteristics of Cases and Controls

There were 202 cases and 202 controls, matched by trainer and age, recruited during the 13-month period. Of these, 103 (51%) were two-year-old horses and 99 (49%) were three years and older. The population characteristics of these horses, general racing and training history, HSEH, and racing career and performance indices, stratified by age are presented in Table 1. Two-year-old horses had a significantly higher number of colts, a longer period of pre-training, and a higher number of days spent on the water and more jump-outs than horses three years and older. Two-year-old horses were more likely to be unraced, had shorter training preparations, fewer days travelling at three-quarter pace and above (faster than 13 m/s; 15 s/furlong; 800 m/min; 48 km/h) and at a gallop (faster than 15 m/s; 13 s/furlong; 900 m/min; 55 km/h), shorter distances travelled at three-quarter pace and above and at a gallop, and shorter distances per event at both three-quarter pace and above and at a gallop. Two-year-old horses accumulated less racing distance, had fewer race starts and lower numbers of wins and first, second or third places than older horses.

### 3.2. Univariable Analysis of Risk Factors for Musculoskeletal Injury

For horses of all ages, the results of univariable analyses of the effect of the population characteristics, the general racing and training history, the HSEH, and the racing career and performance indices on MSI are presented in Table 2. Analyses were repeated for two-year-old horses and for horses three years and older. Insufficient sample size prevented analysis of the effect of racing career and performance indices on MSI in two-year-old horses, as only 13 cases and 14 controls had raced. There was no evidence of strong correlations between most variables, with Pearson’s pairwise correlation coefficients all <0.80 except for (1) days travelled at three-quarter pace and above (faster than 13 m/s; 15 s/furlong; 800 m/min; 48 km/h) and distance travelled at three-quarter pace and above and (2) combined distance travelled during jump-outs, trials and races and distance travelled at a gallop (faster than 15 m/s; 13 s/furlong; 900 m/min; 55 km/h). Interestingly, there was a poor correlation between the number of days worked at a slow pace and the total preparation length (r = −0.01).

### 3.3. Multivariable Analysis of Risk Factors for Musculoskeletal Injury

For horses of all ages, the final multivariable model included five main effect terms: dam parity, total preparation length, days worked at a slow pace (less than 13 m/s; 15 s/furlong; 800 m/min; 48 km/h), the distance travelled at three-quarter pace and above (greater than 13 m/s; 15 s/furlong; 800 m/min; 48 km/h), and the total distance galloped (faster than 15 m/s; 13 s/furlong; 900 m/min; 55 km/h) in the last four weeks (Table 3). All other terms were dropped from the model during the model building process. Post-estimation model checking did not identify any substantial multi-collinearity or model misspecification. Increasing dam parity significantly decreased the odds of MSI, with horses born to mares that have had 5–14 foals having significantly lower odds of MSI than horses born to primiparous mares (odds ratio (OR) 0.27; 95% confidence interval (CI) 0.12, 0.64; *p* = 0.003). Increasing total preparation length also significantly increased the odds of MSI. Horses that had a total preparation length of 15–48 weeks had 5.56 (95% CI 1.59, 19.46; *p* = 0.01) times the odds of MSI than horses that had a total preparation length of 2–7 weeks. The odds of MSI decreased significantly with increasing number of days worked at a slow pace. Horses that worked 18–24 days in the last four weeks at a slow pace had 0.09 times the odds (95% CI 0.03, 0.28; *p* < 0.001) of MSI than horses that worked 0–12 days at a slow pace. The effect of high-speed exercise history on the odds of MSI is illustrated in Figure 1. The odds of MSI decreased with increasing distance travelled at three-quarter pace and above in the last four weeks. Horses that travelled 6.4–12.5 km (32–62 furlongs) at three-quarter pace and above had 0.06 (95% CI 0.01, 0.26; *p* < 0.001) times the odds of MSI than horses that travelled 0–2.8 km (0–14 furlongs) at three-quarter pace and above. The odds of MSI were significantly higher for horses that travelled a total distance of 2.4–3.8 km (12–19 furlongs) at a gallop in the last four weeks than those that travelled between 0 and 1 km (0–5 furlongs) (OR 3.85; 95% CI 1.04, 14.25; *p* = 0.04).

For two-year-old horses, the final multivariable model included four main effect terms: dam parity, total preparation length, days worked at a slow pace, and the distance travelled at three-quarter pace and above (Table 3). Increasing dam parity significantly decreased the odds of MSI (parity 5–14 compared to parity 1, OR 0.08; 95% CI 0.02, 0.36; *p* < 0.001). Increasing total preparation length also increased the odds of MSI (10–14 weeks compared to 2–7 weeks, OR 8.05; 95% CI 1.92, 33.76; *p* = 0.004). The odds of MSI also decreased with increasing number of days worked at a slow pace (18–24 days compared to 0–12 days, OR 0.14; 95% CI 0.03, 0.65; *p* = 0.01). Increasing distance travelled at three-quarter pace and above also decreased the odds of MSI (6.4–12.5 km, 32–62 furlongs compared to 0–2.8 km, 0–14 furlongs, OR 0.13; 95% CI 0.02, 0.91; *p* = 0.04) (Figure 1).

For horses three years and older, the final multivariable model included two main effect terms: days worked at a slow pace and the distance travelled at three-quarter pace and above (Table 3). Increasing number of days worked at a slow pace significantly reduced the odds of MSI (18–24 days compared to 0–12 days, OR 0.08; 95% CI 0.02, 0.37; *p* = 0.001). Horses that travelled 3.0–4.8 km (15–24 furlongs) at three-quarter pace and above had higher odds of MSI than those that travelled 0–2.8 km (0–14 furlongs) (OR 10.02; 95% CI 1.34, 75.77; *p* = 0.03). The odds of MSI then decreased as the distance travelled at three-quarter pace and above (Figure 1).

### 3.4. Analysis by Type of Musculoskeletal Injury

For horses of all ages, the results of univariable analyses of the effect of dam parity, the distance travelled at three-quarter pace and above, and the distance travelled at a gallop in the past four weeks on MSI are presented in detail in Table A1 and summarized in Figure A4 and Figure A5, respectively.

## 4. Discussion

We investigated the risk factors for MSI in Thoroughbred racehorses in South-East Queensland, Australia overall, in two-year-old and older horses and by injury types. Of the population characteristics, increasing dam parity significantly reduced the odds of MSI in horses of all age groups because of the effect in two-year-old horses. Dam parity is more likely to be associated with risk of MSI in two-year-old horses rather than older horses because of the tissue development and adaptation occurring from training [59,60,61]. It is plausible that increasing dam parity could decrease the odds of MSI in two-year-old horses through increasing birthweight and, therefore, increasing volumetric bone mineral density. Multiparous mares are known to produce foals with heavier birthweight than primiparous mares [62]. Heavier bodyweight is associated with a higher volumetric bone density in human children [63], although this information is not available for horses. Increased volumetric bone density is protective for MSI in humans [64] and horses [65,66]. However, our findings were contradictory to a previous study that reported two-year-old horses from multiparous mares had a higher risk of fracture than those from primiparous mares [67]. The difference between the findings may be due to the type of injury. Dam parity may affect the risk of fracture differently to dorsal metacarpal disease, the predominant injury type in this population of two-year-old horses. We observed that the odds of dorsal metacarpal disease decreased with increasing dam parity to a greater extent than for other types of injuries, although this was not statistically significant. Alternatively, the difference in findings may be because this study was a case-control study, and therefore, did not account for time at risk of injury. Further research into the potential mechanisms for how dam parity may affect the risk of MSI is required, as this may be a useful parameter to easily identify horses at increased risk of MSI.

Of the general racing and training history, this study found that a long preparation time (a longer period in which the horse is remaining in race training without rest) was associated with increased odds of MSI in two-year-old, but not older, horses. Two-year-old horses are likely more at risk of MSI as total preparation length increases because they need the rest period to enable their tissues to repair and adapt to the effects of race training. In contrast, tissue adaptation has already occurred in older horses [68] and they can withstand longer periods in race training. Increasing number of days exercised at a slow pace also decreased the odds of MSI in horses of all ages. This is consistent with other reports and our understanding of response to exercise, as the magnitude of the forces and strains generated during slow work are less likely to result in tissue failure than those forces experienced during high-speed exercise [24,25,26]. The protective effect of the number of days worked at a slow pace is not simply due to horses being in the early stages of the preparation and not undertaking high-speed exercise, due to the poor correlation between the number of days worked at a slow pace and the total length of preparation. It is likely that trainers were using slow days to decrease the intensity of training for individual horses that they perceived were at increased risk of injury or not coping with the demands of training, rather than ending their training preparation. Investigating this possibility was beyond the scope of this study. Decreasing the intensity of a training program may, in some situations, be more beneficial than resting the horse. When bone loading ceases with rest, remodelling occurs and bone strength is substantially reduced due to increased osteoclastic activity [24,69]. If remodelling is not complete, bone strength is reduced as the osteoclastic activity has weakened the structure prior to deposition of replacement bone during the osteoblastic phase [24,69]. Returning from rest was strongly associated with the risk of catastrophic humeral fractures [70].

Of the high-speed exercise parameters, the distance travelled at three-quarter pace and above (faster than 13 m/s; 15 s/furlong; 800 m/min; 48 km/h) and the total distance travelled at a gallop (faster than 15 m/s; 13 s/furlong; 900 m/min; 55 km/h) in the past four weeks significantly affected the odds of MSI. The odds of MSI decreased as the distance travelled at three-quarter paced and above increased. Horses of all ages that travelled 6.4–12.5 km (32–62 furlongs) of three-quarter pace and above had 94% lower odds of MSI than those that travelled 0–2.8 km (0–14 furlongs). Two-year-old horses that travelled 6.4–12.5 km (32–62 furlongs) of three-quarter pace and above had 87% lower odds of MSI than those that travelled 0–2.8 km (0–14 furlongs). We attributed these findings to a combination of survival bias and tissue adaptation due to training [11,71]. These findings are biologically plausible because high-speed exercise is required for the bone [24,37,69,72,73,74,75,76] and tendon or ligament [60,77,78,79,80,81,82,83] adaptation necessary to prevent injury. These findings are also consistent with previous studies reporting a decreased risk of MSI with increasing HSEH [17,29,37,38,39,40,41,42,43].

The total distance galloped (trackwork, jump-outs, official trials, and races) in the past four weeks increased and subsequently decreased the risk of MSI for horses of all ages in this study, although this effect was not statistically significant for the individual age categories. Horses that travelled a total distance of 2.4–3.8 km (12–19 furlongs) at a gallop in the last four weeks had four times, and horses that travelled 4.0–6.6 km (20–33 furlongs) had two times, the odds of MSI compared to horses that travelled 0–1 km (0–5 furlongs). This same non-linear pattern was seen when the total distance travelled at three-quarter pace and above were analysed separately within each individual age category. Horses three years and older that travelled 15–24 furlongs at three-quarter pace and above had 10 times the odds of MSI than horses that travelled 0–2.8 km (0–14 furlongs). It is consistent with other reports and biologically plausible that any increase in the high-speed exercise distance travelled would increase the odds of MSI relative to the lowest category of distance travelled, due to the largest forces and strains incurred at the fastest speeds [24,25,26]. However, the increase in odds of MSI with increasing distance galloped followed by a subsequent decrease with further accumulated distance supports the theory that a certain level of galloping is required to enable the tissue adaptation necessary to withstand injury. Once this level has been reached, the risk of injury decreases with further high-speed exercise, up to a point at which further high-speed exercise causes injury [24,37,69,72,73,74,75,76]. The lower odds of MSI for the highest category of distance galloped is also consistent with the effect of survival bias. These findings are also consistent with other studies that reported a curvilinear effect on the risk of MSI [17,36,38,40].

In the current study, the risk factors for MSI in two-year-old horses were equivalent to those for older horses, with the exception of dam parity and the length of the training preparation. The maturity of a racehorse and the ideal starting time for training is strongly debated, with limited supporting literature. Mason et al. (1973) reported a relationship between unsoundness and open distal radial epiphyses, however, there has been no further research to support this study. Further, Rogers et al. conducted a series of experimental studies providing evidence that early race training is beneficial to racehorses, facilitating superior tissue adaptation [59,60,61,68,71]. This is supported by a later study that associated longevity of racing career with starting as a two-year-old [84].

Unfortunately, we were unable to collect data for when the older horses started their race training. We found no association between racing as a two-year-old (or the number of starts as a two-year-old) and injury in older horses, however, we do not know the training history for those horses. Horses may have undertaken a high training volume as two-year-old horses yet had their first race start early in their third year, which would bring our observed effect size towards the null. While the lack of significant findings supports the possibility that there is no long-term harm in racing horses at two years of age, it does not confirm that early exercise is beneficial either. Furthermore, there is the issue of survival bias, whereby those individuals that are injured early are removed and those remaining are at a reduced risk of injury. There is a need for prospective studies to evaluate whether starting these horses at two years of age is associated with an increased or decreased risk of injury.

The effect of HSEH on MSI was also analysed according to type of injury. Unfortunately, caution should be used when interpreting the findings from this part of the analysis due to the wide confidence intervals that reflect the small sample size of sub-categories. This reduced the power to a level where multivariable analysis was not possible, however, prolonging data collection to increase the number of cases in each sub-category was beyond the resources of this study. However, some important patterns of how HSEH affected the odds of different injury types were observed. Firstly, for all types of injury except for those classified as “other” (traumatic disease, feet problems, and developmental orthopaedic disease), the odds of MSI initially increased with increasing HSEH and subsequently decreased. This is consistent with the overall effect of HSEH on the risk of MSI in this study and suggests that a similar mechanism exists for all types of injuries. Interestingly, the peak odds of increasing distance at the gallop for dorsal metacarpal disease was up to three times higher than the peak odds of other types of injuries. It is unclear whether this is due to the type of injury or reflects the higher risk of MSI in two-year-old horses, as dorsal metacarpal disease was seen exclusively in two-year-old horses and a small number of older horses that commenced training at a later age. Further research into the different types of injuries in the same population of horses would confirm or refute the patterns observed in this study.

The main strength of this study was detailed, high-quality data for a large number of risk factors resulting from unique access to a large number of the trainers through personal interviews. Personal interviews ensured that the data collected was both complete and accurate, rather than relying on trainers to complete standardized questionnaires [51,85,86]. The prospective weekly data collection minimised the inherent recall bias of case-control studies [87]. The main limitation of this study is that analysing by age groups and subtypes of injury reduced our statistical power. Additionally, controls were not screened for previous injury, nor followed up for future injury, which could potentially bring our observed effect size towards the null. Furthermore, this study represents a subset of the Australian racing industry and our results may not be globally applicable.

## 5. Conclusions

We identified horses in this population that have particularly high odds of MSI. Two-year-old horses from primiparous mares are at increased odds of MSI, particularly dorsal metacarpal disease. Two-year-old horses that have had a total preparation length of between 10 and 14 weeks also have increased odds of MSI. Horses of all ages that travelled a total distance of 2.4–3.8 km (12–19 furlongs) at a gallop in the last four weeks and horses three years and older that travelled 3.0–4.8 km (15–24 furlongs) at three-quarter pace and above also have increased odds of MSI. We recommend that these horses should be monitored closely for impending signs of MSI. There was also a non-linear relationship between HSEH and MSI, demonstrating the importance of high-speed exercise to enable tissue adaptation to training. This appears to apply to nearly all types of injuries. Finally, in some situations, increasing the number of days worked at a slow pace may be more effective for preventing MSI, if horses are perceived to be at a higher risk, than resting the horse altogether. Early identification of horses at increased risk of MSI and appropriate intervention could substantially reduce the impact of MSI in Thoroughbred racehorses.

## Figures and Tables

**Figure 1 animals-11-00270-f001:**
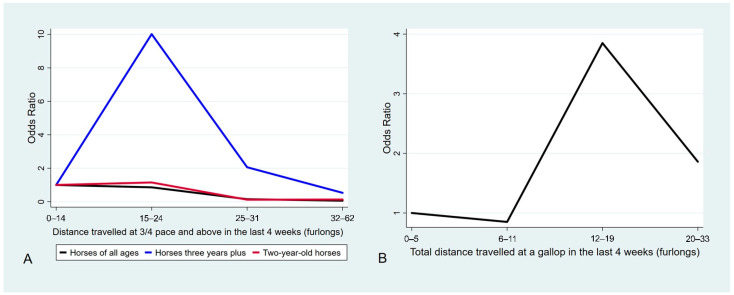
The effect of high-speed exercise history on musculoskeletal injury in Thoroughbred racehorses in South-East Queensland, Australia. (**A**) The effect of increasing distance travelled at three-quarter pace and above (faster than 13 m/s; 15 s/furlong; 800 m/min; 48 km/h) on musculoskeletal injuries (MSI) in horses of all ages, two-year-old horses and horses three years and older. (**B**) The effect of increasing distance travelled at a gallop (faster than 15 m/s; 13 s/furlong; 900 m/min; 55 km/h) on MSI in horses of all ages. Results shown are from multivariable conditional logistic regression analyses for 202 cases of musculoskeletal injury and 202 controls matched by trainer and age. Significance was set at α = 0.05.

**Table 1 animals-11-00270-t001:** The population characteristics, training characteristics, training and racing high speed exercise history, and racing career and performance indices for 202 cases of musculoskeletal injury and their controls (matched by trainer and age), stratified by age.

Variable	Horses of All Ages		Two-Year-Old Horses		Horses Three Years and Older		
	Cases (*n* = 202)	Controls (*n* = 202)	Cases (*n* = 103)	Controls (*n* = 103)	Cases (*n* = 99)	Controls (*n* = 99)	
Population characteristics							
	Median (IQR)	Median (IQR)	Median (IQR)	Median (IQR)	Median (IQR)	Median (IQR)	*p*-value ^#^
Dam Age (years)	9 (7–12)	10 (8–12)	9 (7–11)	10 (8–12)	9 (7–13)	10 (7–12)	0.24
Dam Parity	3 (2–5)	4 (2–6)	3 (2–4)	4 (2–6)	4 (2–6)	3 (2–6)	0.52
	N (%)	N (%)	N (%)	N (%)	N (%)	N (%)	*p*-value ^#^
Sex							<0.001
Entire males	34 (17.0)	34 (17)	31 (30)	33 (32)	3 (3)	1 (1)	
Geldings	82 (40)	88 (43)	24 (23)	23 (22)	58 (59)	65 (66)	
Females	86 (43)	80 (40)	48 (47)	47 (46)	38 (38)	33 (33)	
General racing and training history							
	N (%)	N (%)	N (%)	N (%)	N (%)	N (%)	*p*-value ^#^
Started in a race							<0.001
Yes	97 (48)	105 (52)	13 (13)	14 (14)	84 (85)	91 (92)	
No	105 (52)	97 (48)	90 (87)	89 (86)	15 (15)	8 (8)	
	Median (IQR)	Median (IQR)	Median (IQR)	Median (IQR)	Median (IQR)	Median (IQR)	*p*–value ^#^
Length of training prep (wks)	9 (6–14)	8 (5–15)	7 (5–10)	6 (5–8)	13 (8–24)	14 (8–20)	<0.001
Length of pre-training (wks)	0 (0–2)	0 (0–2)	2 (0–3)	0 (0–3)	0 (0–0)	0 (0–0)	<0.001
Total prep length (wks)	10 (8–15)	10 (7–15)	9 (7–11)	8 (6–10)	14 (8–24)	14 (9–20)	<0.001
Days slow last 4 wks	15 (13–17)	16 (15–18)	15 (13–17)	17 (15–19)	15 (13–17)	16 (15–17)	0.06
Days water-walker last 4 wks	0 (0–0)	0 (0–0)	0 (0–0)	0 (0–0)	0 (0–0)	0 (0–0)	0.01
Days treadmill last 4 wks	0 (0–0)	0 (0–0)	0 (0–0)	0 (0–0)	0 (0–0)	0 (0–0)	0.24
High-speed exercise history							
	Median (IQR)	Median (IQR)	Median (IQR)	Median (IQR)	Median (IQR)	Median (IQR)	*p*-value ^#^
Days 3/4 pace and above last 4 wks	6 (4–7)	7 (5–8)	6 (4–7)	6 (3–7)	6 (5–8)	7 (6–8)	<0.001
Dist 3/4 pace and above last 4 wks (f)	24 (15–31)	27 (16–34)	18 (10–24)	20 (9–26)	30 (22–37)	33 (29–40)	<0.001
Dist 3/4 pace and above per event (f)	4 (3–5)	4 (3–5)	3 (3–4)	3 (3–4)	5 (4–5)	5 (4–5)	<0.001
Days track gallop last 4 wks	5 (2–6)	5 (1–6)	3 (1–5)	3 (0–5)	6 (4–7)	6 (5–7)	<0.001
Dist track gallop last 4 wks (f)	12 (4–16)	12 (2–18)	6 (2–12)	5 (0–12)	16 (11–19)	18 (12–21)	<0.001
Dist track gallop per event (f)	2 (2–3)	2 (1–3)	2 (1–2)	2 (0–2)	3 (2–3)	3 (2–3)	<0.001
Dist jump-outs last 4 wks (f)	0 (0–2)	0 (0–0)	0 (0–2)	0 (0–2)	0 (0–0)	0 (0–0)	<0.001
Dist trials last 4 wks (f)	0 (0–0)	0 (0–0)	0 (0–0)	0 (0–0)	0 (0–0)	0 (0–0)	0.07
Dist races last 4 wks (f)	0 (0–5)	0 (0–6)	0 (0–0)	0 (0–0)	4 (0–7)	5 (0–11)	<0.001
Dist jump-outs, trials and races (f)	4 (0–8)	4 (0–10)	2 (0–5)	0 (0–4)	6 (0–10)	7 (0–12)	<0.001
Total dist galloped (track, jump-outs, trials, races)	16 (4–22)	16 (2–28)	8 (3–17)	6 (0–17)	21 (16–30)	24 (12–33)	<0.001
Racing career and performance indices							
	N (%)	N (%)	N (%)	N (%)	N (%)	N (%)	*p*-value ^#^
Started as two-year-old							
Yes	48 (49)	56 (53)	†	†	35 (42)	43 (47)	†
No	49 (51)	49 (47)	†	†	49 (58)	48 (53)	†
	Median (IQR)	Median (IQR)	Median (IQR)	Median (IQR)	Median (IQR)	Median (IQR)	*p*-value ^#^
No. starts as two-year-old	2 (1–4)	3 (2–5)	†	†	2 (1–4)	2 (2–5)	†
Age at first race (years)	3 (2–3)	2 (2–3)	†	†	3 (2–3)	3 (2–3)	†
Length of career to date (months)	17 (8–29)	17 (8–27)	3 (1–3)	5 (3–8)	19 (12–30)	20 (10–29)	<0.001
No. of starts	10 (3–19)	9 (4–20)	2 (2–3)	3 (2–5)	11 (5–22)	11 (6–21)	<0.001
Starts per 12 months over career	8 (5–11)	8 (6–10)	12 (8–12)	8 (6–10)	8 (5–10)	8 (6–10)	0.11
No. of races won	1 (0–3)	2 (0–3)	1 (0–1)	0 (0–1)	2 (0–4)	2 (1–4)	<0.001
% of races won	17 (0–25)	15 (0–24)	50 (0–50)	0 (0–25)	16 (0–21)	15 (7–24)	0.30
No. of races 1st, 2nd or 3rd place	3 (2–9)	4 (2–9)	2 (1–2)	1 (0–2)	4 (2–9)	6 (2–10)	<0.001
% of races 1st, 2nd or 3rd place	44 (33–55)	46 (31–58)	67 (50–100)	50 (0–71)	43 (30–53)	45 (33–56)	0.11
Total dist raced (5 f/1 km)	11 (3–26)	11 (5–26)	2 (2–3)	3 (2–5)	14 (6–28)	14 (7–29)	<0.001
Average dist per start (f)	6 (6–7)	6 (6–7)	5 (5–6)	5 (5–6)	6 (6–7)	6 (6–7)	<0.001
Total prizemoney ($1000 AUD)	33 (10–83)	41 (12–89)	26 (10–51)	13 (2–67)	34 (10–99)	46 (16–90)	0.03
Prizemoney per start ($1000 AUD)	4 (1–6)	4 (2–8)	13 (3–25)	5 (0–18)	3 (1–5)	4 (2–7)	0.06

IQR = interquartile range; ^#^
*p*-value for comparisons between two-year-old horses and horses three years and older; † Not applicable, Prep = preparation, Wks = weeks, Dist = distance, f = furlongs, No. = number.

**Table 2 animals-11-00270-t002:** The effect of population characteristics, general racing and training history, high-speed exercise history, and racing and performance indices on musculoskeletal injuries in Thoroughbred racehorses in South-East Queensland, Australia.

Variable	Horses of All Ages	*p*-Value ^†^	Two-Year-Old Horses	*p*-Value ^†^	Horses Three Years and Older	*p*-Value ^†^
	Univariable OR (95% CI)		Univariable OR (95% CI)		Univariable OR (95% CI)	
Population characteristics						
Dam Age (years)		0.65		0.28		0.52
4–6	reference		reference		reference	
7–8	1.09 (0.55, 2.16)	0.81	1.82 (0.59, 5.55)	0.30	0.88 (0.35, 2.21)	0.79
9–11	0.79 (0.44, 1.42)	0.42	0.80 (0.36, 1.79)	0.59	0.74 (0.30, 1.79)	0.50
12–27	0.98 (0.52, 1.84)	0.94	0.72 (0.29, 1.79)	0.48	1.27 (0.52, 3.12)	0.60
Dam Parity		0.13		0.02		0.89
1	reference		reference		reference	
2–3	0.56 (0.30, 1.03)	0.06	0.30 (0.12, 0.77)	0.01	1.11 (0.45, 2.73)	0.82
4	0.65 (0.32, 1.32)	0.23	0.28 (0.09, 0.84)	0.02	1.44 (0.53, 3.94)	0.48
5-14	0.47 (0.25, 0.90)	0.02	0.19 (0.07, 0.52)	0.001	1.18 (0.46, 2.99)	0.73
Sex		0.76		0.95		0.45
Entire males	reference		reference		reference	
Geldings	0.91 (0.48, 1.71)	0.76	1.11 (0.52, 2.34)	0.79	0.33 (0.03, 3.20)	0.34
Females	1.09 (0.58, 2.02)	0.79	1.09 (0.56, 2.15)	0.79	0.44 (0.04, 4.61)	0.49
General racing and training history						
Started in a race		0.17		0.82		0.08
Yes	reference		reference		reference	
No	1.62 (0.81, 3.23)	0.17	1.11 (0.45, 2.73)	0.82	2.75 (0.88, 8.64)	0.08
Total preparation length (weeks)		0.15		0.07		0.47
2–7	reference		reference		reference	
8–9	2.04 (1.00, 4.14)	0.05	1.96 (0.77, 5.02)	0.16	2.23 (0.72, 6.88)	0.16
10–14	1.91 (1.02, 3.60)	0.05	3.34 (1.35, 8.24)	0.01	1.03 (0.40, 2.68)	0.95
15–48	1.59 (0.75, 3.34)	0.22	1.50 (0.44, 5.13)	0.52	1.25 (0.45, 3.48)	0.66
Days slow last 4 weeks		<0.001		<0.001		0.01
0–12	reference		reference		reference	
13–15	0.45 (0.21, 0.99)	0.05	0.92 (0.31, 2.69)	0.88	0.22 (0.06, 0.76)	0.02
16–17	0.13 (0.06, 0.32)	<0.001	0.19 (0.06, 0.62)	0.01	0.09 (0.02, 0.35)	0.001
18–24	0.13 (0.05, 0.33)	<0.001	0.15 (0.04, 0.56)	<0.001	0.11 (0.03, 0.45)	0.002
Days water-walker last 4 weeks		0.41		0.37		1.00
0–1	reference		reference		reference	
2–24	0.63 (0.20, 1.91)	0.41	0.57 (0.17, 1.95)	0.37	1.00 (0.06, 15.99)	1.00
Days treadmill last 4 weeks		0.42		0.66		
0–1	reference		reference			
2–12	0.50 (0.09, 2.73)	0.42	0.67 (0.11, 3.99)	0.66		
High-speed exercise history						
Days 3/4 pace and above last 4 weeks		0.02		0.10		0.09
0–3	reference		reference		reference	
4–5	1.99 (0.95, 4.16)	0.07	1.93 (0.79, 4.69)	0.15	2.82 (0.65, 12.33)	0.17
6–7	1.46 (0.75, 2.84)	0.27	2.22 (0.92, 5.37)	0.08	0.82 (0.28, 2.41)	0.71
8–12	0.65 (0.32, 1.31)	0.23	0.69 (0.23, 2.05)	0.50	0.52 (0.18, 1.48)	0.22
Distance 3/4 pace and above last 4 weeks (furlongs)		<0.001		0.09		0.01
0–14	reference		reference		reference	
15–24	1.84 (0.97, 3.48)	0.06	1.53 (0.77, 3.03)	0.22	5.52 (0.89, 34.21)	0.07
25–31	0.66 (0.31, 1.39)	0.27	0.40 (0.13, 1.20)	0.10	1.41 (0.43, 4.68)	0.57
32–62	0.33 (0.15, 0.72)	0.01	0.61 (0.15, 2.41)	0.48	0.41 (0.15, 1.17)	0.10
Distance 3/4 pace and above per event (furlongs)		0.35		0.62		0.79
0–2	reference		reference		reference	
3	1.35 (0.72, 2.54)	0.35	1.24 (0.61, 2.53)	0.55	1.55 (0.34, 6.99)	0.57
4–5	1.15 (0.56, 2.37)	0.71	2.03 (0.71, 5.84)	0.71	0.94 (0.25, 3.49)	0.92
6–11	0.66 (0.27, 1.59)	0.35			0.86 (0.22, 3.28)	0.82
Days track gallop last 4 weeks		0.31		0.32		0.41
0–1	reference		reference		reference	
2–4	1.62 (0.90, 2.89)	0.11	1.81 (0.91, 3.62)	0.09	1.31 (0.42, 4.07)	0.64
5–6	1.17 (0.55, 2.47)	0.68	1.08 (0.38, 3.03)	0.89	1.17 (0.36, 3.77)	0.79
7–9	0.97 (0.51, 1.85)	0.94	1.77 (0.65, 4.82)	0.27	0.65 (0.25, 1.65)	0.36
Distance track gallop last 4 weeks (furlongs)		0.04		0.53		0.05
0–5	reference		reference		reference	
6–11	1.40 (0.73, 2.68)	0.31	1.71 (0.80, 3.67)	0.17	0.77 (0.21, 2.84)	0.69
12–19	1.49 (0.79, 2.79)	0.22	1.42 (0.59, 3.41)	0.43	1.29 (0.49, 3.41)	0.60
20–33	0.57 (0.26, 1.25)	0.16	0.91 (0.14, 6.07)	0.93	0.45 (0.16, 1.24)	0.12
Distance track gallop per event (furlongs)		0.50		0.66		0.70
0–1.4	reference		reference		reference	
1.5–2	1.53 (0.80, 2.93)	0.20	1.64 (0.75, 3.56)	0.21	1.25 (0.38, 4.11)	0.71
2.1–2.9	1.33 (0.70, 2.52)	0.39	1.34 (0.54, 3.31)	0.53	1.15 (0.43, 3.09)	0.78
3–10.6	1.03 (0.53, 2.02)	0.92	1.40 (0.52, 3.81)	0.51	0.78 (0.28, 2.17)	0.64
Distance jump-outs last 4 weeks (furlongs)		0.07		0.12		0.37
0–1	reference		reference		reference	
2–9.3	1.75 (0.95, 3.23)	0.07	1.75 (0.86, 3.56)	0.12	1.75 (0.51, 5.98)	0.37
Distance trials last 4 weeks (furlongs)		0.57		0.28		0.85
0–1	reference		reference		reference	
2–14.2	1.17 (0.67, 2.05)	0.57	1.63 (0.67, 3.92)	0.28	0.93 (0.45, 1.93)	0.85
Distance races last 4 weeks (furlongs)		0.78		0.78		0.33
0–1	reference		reference		reference	
2–22.2	0.86 (0.29, 2.55)	0.78	0.86 (0.29, 2.55)	0.78	0.73 (0.38, 1.38)	0.33
Combined distance jump-outs, trials and races (furlongs)		0.01		0.08		0.09
0–3.9	reference		reference		reference	
4–9.9	1.84 (1.03, 3.28)	0.04	2.30 (0.93, 5.68)	0.07	1.61 (0.74, 3.50)	0.23
10–22.2	0.72 (0.38, 1.36)	0.31	0.60 (0.20, 1.82)	0.37	0.73 (0.32, 1.64)	0.44
Total distance galloped (track, jump-outs, trials, races)		0.04		0.30		0.70
0–5	reference		reference		reference	
6–11	1.03 (0.50, 2.15)	0.93	1.31 (0.57, 3.09)	0.53	0.48 (0.10, 2.36)	0.37
12–19	2.34 (1.14, 4.78)	0.02	2.00 (0.83, 4.84)	0.12	2.71 (0.77, 9.53)	0.12
20–33	0.94 (0.50, 1.77)	0.86	1.40 (0.54, 3.61)	0.49	0.70 (0.28, 1.76)	0.45
Racing career and performance indices						
Number starts as two-year-old		0.50				0.60
0	reference				reference	
1–2	0.85 (0.40, 1.81)	0.68			0.78 (0.36, 1.68)	0.52
3–5	0.45 (0.16, 1.26)	0.13			0.50 (0.18, 1.41)	0.19
6–9	0.65 (0.16, 2.54)	0.53			0.82 (0.20, 3.41)	0.78
Age at first race (years)		0.19				0.19
2	reference				reference	
3	1.31 (0.68, 2.52)	0.41			1.31 (0.68, 2.52)	0.41
4	3.61 (0.90, 14.42)	0.07			3.61 (0.90, 14.42)	0.07
Length of career to date (months)		0.99				0.85
1–7	reference				reference	
8–16	1.16 (0.40, 3.33)	0.79			1.68 (0.52, 5.39)	0.39
17–28	1.06 (0.35, 3.21)	0.92			1.33 (0.42, 4.20)	0.63
29–68	1.19 (0.33, 4.31)	0.79			1.48 (0.39, 5.62)	0.58
Number of starts		0.62				0.70
1–3	reference				reference	
4–8	0.72 (0.24, 2.20)	0.57			0.99 (0.30, 3.30)	0.99
9–19	0.81 (0.24, 2.76)	0.73			1.01 (0.28, 3.61)	0.98
20–67	0.43 (0.10, 1.93)	0.27			0.55 (0.12, 2.58)	0.45
Starts per 12 months over career duration		0.46				0.44
0.4–5.7	reference				reference	
5.8–8.2	0.46 (0.18, 1.21)	0.12			0.45 (0.16, 1.23)	0.12
8.3–10	0.73 (0.29, 1.80)	0.49			0.72 (0.29, 1.81)	0.49
10.1–24	0.69 (0.26, 1.81)	0.49			0.83 (0.31, 2.24)	0.72
Number of races won		0.72				0.68
0	reference				reference	
1–2	0.57 (0.22, 1.48)	0.25			0.53 (0.19, 1.47)	0.22
3–5	0.63 (0.20, 2.00)	0.44			0.60 (0.18, 1.96)	0.39
6–13	0.62 (0.13, 2.96)	0.55			0.58 (0.12, 2.87)	0.51
% of races won		0.26				0.19
0	reference				reference	
1–15	0.42 (0.14, 1.29)	0.14			0.44 (0.14, 1.40)	0.17
16–25	0.84 (0.28, 2.49)	0.76			0.93 (0.29, 2.96)	0.90
26–100	0.63 (0.21, 1.84)	0.40			0.52 (0.16, 1.67)	0.28
Number of races 1st, 2nd or 3rd place		0.49				0.47
0	reference				reference	
1–2	1.86 (0.50, 6.92)	0.36			1.88 (0.50, 7.06)	0.35
3–6	1.08 (0.25, 4.62)	0.91			1.04 (0.24, 4.50)	0.96
7–27	0.87 (0.19, 4.02)	0.86			0.85 (0.18, 3.95)	0.83
% of races 1st, 2nd or 3rd place		0.50				0.38
0	reference				reference	
1–30	2.40 (0.56, 10.20)	0.24			2.70 (0.62, 11.86)	0.19
31–50	1.19 (0.30, 4.65)	0.81			1.17 (0.30, 4.63)	0.82
51–100	1.29 (0.34, 4.97)	0.71			1.21 (0.31, 4.69)	0.78
Total distance raced (5 furlongs/1 km)		0.87				0.89
1–4.3	reference				reference	
4.4–11.2	0.81 (0.27, 2.44)	0.70			1.13 (0.34, 3.74)	0.84
11.3–26.5	0.66 (0.19, 2.21)	0.50			0.84 (0.24, 2.97)	0.79
26.6–139.9	0.53 (0.12, 2.40)	0.41			0.69 (0.15, 3.27)	0.64
Average distance per start (furlongs)		0.13				0.15
4.5–5.6	reference				reference	
5.7–6.0	0.41 (0.13, 1.27)	0.12			0.53 (0.16, 1.69)	0.28
6.1–6.8	1.00 (0.27, 3.63)	0.99			1.39 (0.35, 5.51)	0.64
6.9–10.4	0.43 (0.12, 1.55)	0.20			0.57 (0.15, 2.13)	0.41
Total prizemoney ($1000 AUD)		0.25				0.40
0–11.0	reference				reference	
11.1–36.9	0.43 (0.13, 1.42)	0.17			0.46 (0.14, 1.54)	0.21
37.0–88.6	0.28 (0.07, 1.02)	0.05			0.34 (0.09, 1.27)	0.11
88.7–1039.5	0.26 (0.06, 1.09)	0.07			0.30 (0.07, 1.27)	0.10
Prizemoney per start ($1000 AUD)		0.05				0.06
0–1.8	reference				reference	
1.9–3.7	0.16 (0.04, 0.62)	0.01			0.16 (0.04, 0.62)	0.01
3.8–6.8	0.25 (0.07, 0.92)	0.04			0.25 (0.07, 0.90)	0.03
6.9–67.2	0.17 (0.04, 0.69)	0.01			0.18 (0.04, 0.74)	0.02

OR = odds ratio, CI = confidence interval. Results from univariable conditional logistic regression analysis for 202 cases of musculoskeletal injury and 202 controls matched by trainer and age. Analyses were then repeated for 103 two-year-old cases of musculoskeletal injury and 103 two-year-old controls and 99 cases of musculoskeletal injury and 99 controls three years and older, matched by age and trainer. Variables eligible for inclusion in multivariable model at α = 0.2. ^†^
*p*-Values reported are for the Wald test of the overall effect of each exposure variable on the odds of musculoskeletal injury and the results of univariable conditional logistic regression within each category of the exposure variable.

**Table 3 animals-11-00270-t003:** The effect of population characteristics, general racing and training history, and high-speed exercise history on musculoskeletal injuries in Thoroughbred racehorses in South-East Queensland, Australia.

Population Characteristics	Horses of All Ages		Two-Year-Old Horses		Horses Three Years and Older	
	Adjusted OR (95% CI)	*p*-Value ^†^	Adjusted OR (95% CI)	*p*-Value ^†^	Adjusted OR (95% CI)	*p*-Value ^†^
Dam Parity		0.01		0.01		
1	reference		reference			
2–3	0.55 (0.25, 1.22)	0.14	0.19 (0.05, 0.70)	0.01		
4	0.62 (0.25, 1.54)	0.30	0.16 (0.03, 0.76)	0.02		
5–14	0.27 (0.12, 0.64)	0.003	0.08 (0.02, 0.36)	0.001		
General racing and training history						
Total preparation length (weeks)		0.01		0.04		
2–7	reference		reference			
8–9	4.77 (1.69, 13.43)	0.003	5.17 (1.27, 21.02)	0.02		
10–14	4.73 (1.65, 13.52)	0.004	8.05 (1.92, 33.76)	0.004		
15–48	5.56 (1.59, 19.46)	0.01	2.61 (0.51, 13.44)	0.25		
Days slow last 4 weeks		<0.001		0.004		0.005
0–12	reference		reference		reference	
13–15	0.31 (0.12, 0.84)	0.02	1.04 (0.27, 4.00)	0.96	0.19 (0.05, 0.74)	0.02
16–17	0.10 (0.03, 0.28)	<0.001	0.20 (0.05, 0.83)	0.03	0.08 (0.02, 0.36)	0.001
18–24	0.09 (0.03, 0.28)	<0.001	0.14 (0.03, 0.65)	0.01	0.08 (0.02, 0.37)	0.001
High-speed exercise history						
Distance 3/4 pace and above last 4 weeks (furlongs)		<0.001		0.05		0.007
0–14	reference		reference		reference	
15–24	0.86 (0.30, 2.52)	0.79	1.15 (0.43, 3.08)	0.78	10.02 (1.34, 75.77)	0.03
25–31	0.15 (0.04, 0.64)	0.01	0.12 (0.02, 0.70)	0.02	2.06 (0.49, 8.60)	0.32
32–62	0.06 (0.01, 0.26)	<0.001	0.13 (0.02, 0.91)	0.04	0.53 (0.14, 1.93)	0.33
Total distance galloped (track, jump-outs, trials, races)		0.08				
0–5	reference					
6–11	0.85 (0.26, 2.77)	0.79				
12–19	3.85 (1.04, 14.25)	0.04				
20–33	1.87 (0.51, 6.82)	0.34				

Results from multivariable conditional logistic regression analysis for 202 cases of musculoskeletal injury and 202 controls matched by trainer and age. Analyses were then repeated for 103 two-year-old cases of musculoskeletal injury and 103 two-year-old controls and 99 cases of musculoskeletal injury and 99 controls matched by age and trainer. Variables were considered significant at α = 0.05. ^†^
*p*-Values reported are for the Wald test of the overall effect of each exposure variable on the odds of musculoskeletal injury and the results of univariable conditional logistic regression within each category of the exposure variable.

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
