# Peer review of "The Risk Factors for Musculoskeletal Injuries in Thoroughbred Racehorses in Queensland, Australia: How These Vary for Two-Year-Old and Older Horses and with Type of Injury"

_animals, 2021, doi:10.3390/ani11020270_

Round 1

Reviewer 1 Report

The presented paper is very interesting and covers a lot of new information on thoroughbred training and musculoskeletan disorders. The language of the paper is clear and understandable as a whole, however the paper should be read again to let the other not specialist readers follow the subject. Some sentences should be simplified, some definitions given.   

Simply summary should be written in a more easy way. It might not be a shortcut from the abstract being the shortcut from the paper. Short names may not be used in abstracts – please change HSEH into the full words. Use number of horses investigated and some information on your methods in the abstract, the scientific one.

Some words are not used in a proper way – the marker can be used as genetic marker or biochemical marker. So something easy recognizable, giving specified information about the whole subject. What kind of the marker do you mind here? Parameters of HSEH is clearer.

All paper must be change to be internationally readable – you may use furlong in brackets, but not as the main word. Please change it through all the paper. Please use SI parameters mainly. You may left all data analysis in furlongs, but please use in material, results and in discussion section SI wording.    

The biomechanical wording is not clear – perhaps you should place special vocabulary in the beginning of your paper. What do you mean by three-quarter pace? It is not a common biomechanical word. Are you sure that your vocabulary through all the paper is correct? Gallop is a four beat gate, faster than canter being three beat gate. Are you sure through all your paper to use gallop, galloping? If you are sure that you mean real gallop you should left it, but please check it.

Perhaps it is correct racing vocabulary, but please check the biomechanical vocabulary that should be used in scientific papers. Perhaps the book “Equine locomotion” by Back and Clayton will be good to look at. In other case your paper will not be understand internationally.

It is not quite clear if the aim 2 was not too close to your current study.

Material and methods are mostly clear and many words are explained here, however it should be given also earlier in vocabulary and used in the proper way. Canter and gallop should be used. Perhaps “slow gaits” should be used for slow pace. Be aware that “pace” is a specific kind of gait, with the foot sequence inadequate to canter or gallop.  Please check it carefully through your paper to be clear in all matters. Not all other term are still clear – what is it jump-out? What is the difference between official trail and race? Please do not use shortcuts for your money, write it in words.

I am not a specialist of the statistical methods used in your paper, however they look proper used.  Please use some citation for such methods. I suppose that you need to have comparable methods with other authors, so you need comparable analysis. However you should be also aware of the statistical limitations you create using such methods. For example – if you base your statistics on simple tests you complicate the solving and its interpretation. In case of all normal and not normal distributed data you may use procedure Glinmixed from the SAS program and receive a lot of comparisons in one testing procedure.  By using simply comparison you are not able for example to investigate the effect of the horse genetics (quality of the horse) that may bias your data and results. The mare parity may be connected with the quality of the stallion used in specific time period (for example horses coming from the oldest/youngest parity could be better stallions from the genetic point of view). Probably next paper should check it.

From the part 3.2 there is no line numbering.

Please specify:

L 229 – known and unknown differences between trainers ? not clear

L 276 – water-water?

L 277 - , fewer days, ? of what?

Please move all diagrams with industry matters into the additional material (except 2A). Think about changing the direction/orientation of tables  - there are a lot of them and it is difficult to follow them within the text. Perhaps % could be below N in brackets in one column? The same about IQR under the Median?

Figure 3 – 1 is after 8 in Odds ratio.  Should be 10 I suppose.

Some aspects are not clear – the correlations that are cited twice should be more introduced in the text – write in the result and discussion part by using them what is it from in some words. It is written about correlations in the method part however as not presented in detail in the result part.

Discussion and conclusions are not numbered – so it is difficult to review in detail.

Some sentences are too long and complicated. Without SI parameters and clear biomechanical vocabulary it is difficult to follow these parts.

In conclusion or perhaps in discussion one aspect is lucking, important one:

“Two –year – old horses having long preparation period have increased odds of MSI.”

It should be mention that they could not be mature enough to be trained so early. When did they started? How is it checked before training that a horse is ready for the training? Are they x-rayed before the start? 24 months minus almost 4 months of preparation it is 20 months. The increased odds of MSI may come from too early start not too long training.

If the sentence will be left that way someone could understand that it is better not to prepare/train a young horse for a long period. And it is not the truth and could not stay that way through the paper. The maturity, start point for training is not without meaning. Please discuss this aspect in the paper.

Reviewer 2 Report

The authors aims to 1) Determine the risk factors for MSI 2) Determine whether the risk factors are different for two-year-old and older horses and 3) Determine whether risk factors vary with type of MSI.

It is not frequent to read a manuscript/study with this level of design, in data collection and analysis.

Some acronyms need to be clarified earlier in the document!

Perhaps the results could  be sorted a bit more, so that reading is friendlier, considering the impact that these results could have over trainers and clinicians.

I was wondering if the authors used or try to use a mixed model approach, using the origin of the trainers as random effect?

There is a certain degree of controversy regarding the use of conditional logistic regression when there is a match for more than one variable, as occurs in some of the analyzes in this study. Have the authors considered this?

Some specific comments can be read in the .pdf attached file
